# Consumer Acceptance and Production of In Vitro Meat: A Review

**Kevin Kantono** [1], **Nazimah Hamid** [1,*], **Maya Murthy Malavalli** [1], **Ye Liu** [1], **Tingting Liu** [1] **and Ali Seyfoddin** [2]

1   Department of Food Science, Auckland University of Technology, Private Bag 92006,
    Auckland 1142, New Zealand; kkantono@aut.ac.nz (K.K.); maya.malavalli@aut.ac.nz (M.M.M.);
    ye.liu@aut.ac.nz (Y.L.); lilyttliu@outlook.com (T.L.)
2   Drug Delivery Research Group, School of Science, Auckland University of Technology,
    Auckland 1010, New Zealand; ali.seyfoddin@aut.ac.nz
*   Correspondence: nhamid@aut.ac.nz

**Abstract:** In vitro meat (IVM) is a recent development in the production of sustainable food. The consumer perception of IVM has a strong impact on the commercial success of IVM. Hence this review examines existing studies related to consumer concerns, acceptance and uncertainty of IVM. This will help create better marketing strategies for IVM-producing companies in the future. In addition, IVM production is described in terms of the types of cells and culture conditions employed. The applications of self-organising, scaffolding, and 3D printing techniques to produce IVM are also discussed. As the conditions for IVM production are controlled and can be manipulated, it will be feasible to produce a chemically safe and disease-free meat with improved consumer acceptance on a sustainable basis.

**Keywords:** consumer perception; in vitro meat; IVM production; cells; culture conditions; self-organising technique; 3D printing

## 1. Introduction

In vitro meat is a type of meat that is produced using animal cells, under laboratory conditions [1,2]. IVM has received global attention and is considered a meat of the future by many advocates. This is because IVM production is more sustainable and less emission intensive than the meat obtained from traditional farming. IVM production will also reduce animal welfare issues such as production of factory farmed animals and brutal slaughtering of animals that are associated with traditional farming. IVM can confer direct benefits to public health, such as reducing the spread of food-borne illness. Despite the environmental and public health benefits, commercialisation of this technology is still facing multiple hurdles. The most significant challenges for IVM are its price, consumer perception, and acceptance of technology used to produce IVM.

Consumer perception of novel food products depends on factors such as effects on health, nutrition, taste, safety, affordability, food preference, personal beliefs, cultural identities, religious views, sensory quality, and nature of the product. Based on these factors, consumers will either accept or reject IVM. A considerable amount of empirical literature has been published on the consumer perception of IVM. Most studies have reported consumer concerns on IVM rather than acceptance. Furthermore, perceptions are varied and ambiguous in some studies. For example, some researchers have stated that IVM will remain a niche market, while other researchers think that IVM is predisposed to consumer reluctance before it can be accepted. The discrepancy in studies on consumer perception of IVM motivated us to perform a systematic review on this topic to provide an understanding of factors that impact IVM acceptance.

IVM works on the principle of cellular agriculture, where stem cells are extracted from donor animals through biopsy. The extracted cells usually belong to embryonic stem cells, adult stem cells, mesenchymal stem cells or induced pluripotent stem cells. These cells are cultured either by scaffolds, self-organising techniques, or in a sterile bioreactor. The extracted cells, under favourable conditions, undergo cell proliferation and differentiation to form myofibers, which eventually form muscle tissues. These muscle tissues combine to form skeletal muscles, which can be harvested as edible meat. Efficient production of IVM is needed for the commercialisation of IVM. In the past decades, researchers have worked relentlessly to increase the IVM production efficiency. Various cells, culture conditions, and processing techniques have been used for IVM production. Hence, the second objective of this review was to evaluate the IVM production conditions and techniques, as well as identify the advantages and gaps in research related to IVM production. This will benefit future research on IVM. In general, the aim of this review was to understand the factors that influenced consumer acceptance of IVM and to identify suitable production methods of IVM that will benefit the commercialisation of IVM production.

## 2. Methodology

This literature analysis was carried out by searching a combination of keywords in the Scopus database and Web of Science. The key word strings and search strategy used are summarised in Table 1. The first search on consumer perception of IVM yielded 230 results, and the second search direction on IVM production yielded 1200 results. The objective of this review was to understand factors that influenced consumer perception of IVM and to describe the current production methods of IVM. The research studies that fit in the two objectives of this review were selected for further analysis. After screening, 136 potential publications were identified and analysed in this review.

**Table 1.** Database and key words used.

| Database | Search Terms Used |
| --- | --- |
| Scopus | First search: "in vitro meat" or "lab grown meat" or "animal free meat" or "cultured meat" And "perception"/"consumer"/"concern"/"acceptance" Second search: "in vitro meat" or "lab grown meat" or "animal free meat" or "cultured meat" And "production"/"process" |
| Web of Science | First search: "in vitro meat" or "lab grown meat" or "animal free meat" or "cultured meat" And "perception"/"consumer"/"concern"/"acceptance" Second search: "in vitro meat" or "lab grown meat" or "animal free meat" or "cultured meat" And "production"/"process" |

## 3. Consumer Perception of IVM

The existing studies on consumer acceptance of IVM have shown polarised findings, with some research reporting good consumer acceptance of IVM, while others described more consumer concerns regarding IVM. These contradictory findings are likely due to the following: consumers' attitudinal differences, consumers' meat attachment, food neophobia, disgust, palatability problems, price concerns, perceived health and safety risks, sociocultural beliefs, lack of familiarity of IVM, animal welfare and environment, ethical concerns on IVM technology and its unnaturalness, possibility of animal extinction, and food inequality.

### 3.1. Consumer Acceptance of IVM

Despite numerous concerns, some consumers continue to support the idea of IVM due to environment concerns, as well as health and animal welfare benefits. Generally,

consumer acceptance of IVM mainly depends on three main factors: sociodemographics, consumers' general views on food technologies, and consumer attitudes towards the environment and animal welfare.

### 3.1.1. Sociodemographics

Demographics is an important predictor of IVM acceptance, which mainly depends on gender, age, education, income, eating habits and political affiliation of potential consumers.

#### Gender

Gender is one of the most significant predictors of consumer acceptance of IVM. Previous research showed that meat eating is commonly seen as a masculine trait. Men preferred "real" meals that included meat because it invoked a feeling that is central to their sense of self [3,4]. Such gender-based acceptance is seen in IVM as well. Previous studies showed that men are more open to IVM compared to women [5–8]. This is likely backed by Western masculinity [3] and eagerness among male consumers to try newer technologies in general [9,10].

Research has also found that women generally had negative attitudes towards meat eating. Consequently, women are more inclined to either reduce meat consumption or follow vegetarianism/veganism [11]. Henceforth, vegetarianism/veganism is considered a feminine trait [4]. On this note, previous studies have shown that women are very sceptical of trying novel food innovations, due to the fear of unknown risks and food safety threats [12]. Similar findings are observed in IVM as well, as women were disinclined to engage with IVM [13]. Conversely, Bryant et al. [14] pointed out an unusually higher acceptance rate among Chinese women. However, this could be attributed to skewed sampling in the study that used young, educated and urban-dwelling Chinese women consumers who were already familiar with IVM

#### Age

Age is another critical factor that determines IVM acceptance. Research showed that young consumers, particularly those under 25 years of age are more receptive to IVM and were more likely to purchase IVM compared to older consumers [15]. This is because youth and education are the early adopter's traits for any new technology [16]. Conversely, research showed that elderly consumers' were less willing to try IVM [4] because they were not open to newer experiences and preferred not to deviate from their accustomed habits [17] like traditional meat consumption.

#### Education

Some researchers have studied the effects of education/qualification levels on IVM acceptance. The literature reports that participants with higher education have a higher willingness to try IVM, particularly the young (under 25), and the educated [11,14] compared with the less educated consumers. These findings are because educated consumers are likely to make rational decisions that are not based on naturalness of the product [18]. On the contrary, Hocquette [19] found that educated consumers believed that "artificial meat will not necessarily reduce animal requirements" or dramatically reduce the carbon footprint of traditional meat production. Thus, these consumers felt that IVM would not serve the purpose. Nevertheless, the reason for these contradictory findings might be due to the differences in the study design. The participants surveyed in this study were traditional meat industry professionals and meat scientists. In addition, the question design was poor, with answer options that are not mutually exclusive for the key measures.

### 3.1.2. Eating Habits
#### Omnivores

Eating habits are another critical predictor of IVM consumer acceptance. Previous research showed that individuals with higher meat consumption were less receptive to

IVM [14,18]. These findings are likely due to higher meat attachment, food neophobia, and usual reluctance to try new foods among consumers. Few studies have shown that some meat-eating consumers considered IVM as a meat substituent for beef. Consequently, such consumers have a higher inclination towards IVM. These findings are in line with, Mancini et al. (2019) who showed that about 57% (n = 575) of meat-eating Italian consumers were willing to try IVM compared to other consumers with different eating habits. Likewise, even Chinese consumers with higher meat consumption rates exhibited higher purchase likelihood [4].

Pescatarian

Pescatarians believed that IVM would be healthy and tasty, indicating that pescatarians may have higher inclination towards IVM [14]. However, Wilks [6] reported that consumers were most unlikely to eat fish if it was an IVM product.

Vegetarian and Vegan

The perception of IVM by vegetarians and vegans are somewhat different, as research showed that vegetarians have a slight inclination towards IVM [18] due to its 'meat substituent' image. However, the rate of willingness to engage in IVM appeared to be the lowest for both vegetarians and vegans [20].

### 3.1.3. Country

The current literature shows that most US consumers were willing to try IVM, but only one third were willing to eat IVM regularly [21]. The authors also reported US consumers' disgust towards IVM. On the other hand, the meat eaters in the US did not indicate a high purchase likelihood [22], which is likely due to higher meat attachment observed in the Western countries. Laestadius [23] further stated that such acceptance could be due to the "already industrialized nature of the US agricultural landscape and food system" compared to Europe.

The initial reactions of consumers from European countries such as the Netherlands, Belgium, Italy, and UK towards IVM were underpinned by feelings of disgust and considerations of unnaturalness [18,24]. However, the consumer acceptance of cultured meat will depend on the product-related expectations and experience of consumers [18,24]. The perception of IVM might change when it becomes more available and affordable. Eurobarometer (2005) reported that 88% (n = 12,369) of Cyprian consumers and 23% (n = 12,369) of Bulgarian consumers expressed their unwillingness to engage with IVM technology. Such reactions are due to the aftermath of a prevalent ban on GMO foods in European countries [11].

Emerging markets such as India and China have shown higher acceptance towards IVM [14]. This could be due to higher meat attachment and lower food neophobia. Furthermore, such acceptance is also due to increased familiarity with IVM among consumers, which reduces the risk of immediate rejection. However, the information on acceptance by Indian and Chinese consumers should be considered with care because it involved a skewed sample [25]. Similar findings have been observed in other Asian countries [14], which could be due to lower meat attachment and higher receptiveness to novel food technologies. In contrast, New Zealand consumers have shown minimal acceptance towards IVM [14]. This could be due to two reasons: firstly, the high levels of meat consumption among New Zealanders; secondly, the nature of the sample population [7] that mainly consisted of elderly and female consumers.

### 3.1.4. Political Affiliation

IVM acceptance and traditional meat consumption have a strong connection with political affiliation. Studies showed that consumers with the right/conservative affiliation are often pro-traditional meat [18]. These consumers believe that IVM would have adverse effects on traditional farming [14], unlike the traditional meat industry. Consumers with left

or liberal political affiliation have a higher acceptance of IVM and perceive IVM as a solution to the global warming issues. In addition, liberals believed that IVM is an ethical, natural and tasty alternative. As a result, they have a higher willingness to eat IVM [26]. These findings have been observed among the US and Indian consumers [18]. The continuous support from the left/liberals was likely due to the progressive and moral thinking towards issues such as animal welfare, environmental impacts and food supply [27].

### 3.1.5. Income

Studies have shown that consumers with higher incomes are more receptive to IVM compared to lower-income groups [23,28]. However, the relationship between higher income and consumer acceptance is unclear. The increased receptiveness towards IVM by higher income groups could be due to higher environmental awareness and analytical thinking among educated consumers, as well as their purchase behaviour. On the contrary, Tucker (2014) argued that neither higher nor lower-income groups but those with mid-range income of $49,000–$110,000 NZD were more receptive to IVM. This could be attributed to higher-income consumers opting for superior quality cuts of traditional meat while the lower-income groups opt for cheap supermarket meat, particularly in countries like New Zealand.

### 3.1.6. Religion

The perception of IVM might be influenced by religion. In Judaism, some rabbis would consider IVM Kosher if the cells come from a Kosher-slaughtered animal [29]. The debate of the Kosher status is ongoing. For a complete discussion about the Kosher status, see the article published by Kenigsberg [30]. In Islam, the IVM is considered halal if the cells used are from a halal-slaughtered animal and the culture medium used is halal [31]. Jewish, Muslims and Buddhist showed less preference over IVM compared to meat [29]. However, Hindus find IVM slightly more appealing compared to what they currently eat [18]. Hindus are likely to consider IVM as a way of avoiding harming animals, and some may decide it is permissible to eat provided that it is not beef [29].

### 3.1.7. The Reasons for Accepting of IVM

Besides sociodemographic factors, such as gender, age, education, eating habits, political affiliation and income, consumers would also be willing to try IVM due to perceived benefits to the environment, animal welfare, food security and benefits.

### Environmental Reasons

Livestock rearing and the traditional meat industry are facing a lot of negative repercussions such as global climate changes, global warming, greenhouse gas emissions, increased carbon footprint and increased water and land usage. On the other hand, IVM production requires 99% less land, 45% less energy, and 96% less greenhouse gas emissions compared to traditional meat industry [20]. Consumers are aware that IVM is environmentally sustainable, as it reduces the emission of greenhouse gases [14,20,28,32]. For these reasons, consumers have expressed their willingness to engage with IVM [2,20,33].

### Animal Welfare

Many researchers claim that IVM is beneficial for animal welfare because it prevents animal suffering as large scale production of IVM employs animal cell culture [14,28] requiring no animals to be slaughtered for meat purposes. Not many consumers feared that IVM may result in the reduction of animals [11,34]. In fact, many believed that IVM will improve the animal welfare conditions [35]. Thus, consumers' acceptance towards IVM is mainly due to its potential benefits on animal welfare and animal ethics [11,33].

Food Security

The United Nations stated that "about 815 million hungry people must be nourished today, and an additional 2 billion people are expected to require nourishment by 2050" [36]. However, this may not be possible with the current traditional meat industry practices because traditional farmed meat in the long run is not sustainable, and land for farming will run out eventually. Global hunger can be reduced by production of sustainable meat alternatives like IVM that can potentially solve global meat demand issues [20]. It has been reported that consumers worry about food inequality and believe that IVM can be used to feed underprivileged consumers [20,23,34]. However, these assumptions on food inequality are unclear, and the focus towards a hypothetical product which is currently commercially unavailable in many countries is not a viable option.

Health Benefits

Despite the initial scepticism about health and nutritional impacts of IVM, some consumers still believed that IVM has potential health and safety benefits such as higher safety, higher quality standards, and reduced risk of bovine diseases and zoonotic infections such as *Escherichia coli* and *Salmonella* infections, which are commonly found with traditional meat consumption [14,20,23,28]. Consumers believed that IVM was healthy because it had minimal fat content [19,32,37], unlike traditional fatty meats, which are linked to cardiovascular diseases. On this note, the majority of consumers felt that IVM was entirely safe [38]. Likewise about 28.6% of consumers in Europe perceived IVM to be healthy [39]. This might be because there are no studies on the adverse health implications of IVM, unlike traditional meat, which has been reported to be result in colon disorders with excessive and prolonged consumption [20,34].

### 3.2. Consumer Uncertainty about IVM

### 3.2.1. Controllability

Consumers are wary of IVM controllability as they worry about "mutations", bacterial contamination, risk management and potential violation in the IVM production process [14]. However, such confusions are mainly due to general scepticism towards novel food innovations and lack of familiarity among potential consumers. These concerns can be resolved as technology advances and become familiar with consumers. Furthermore, these concerns will decline once the public is aware of the food safety regulations put forth by prospective food companies.

### 3.2.2. Feasibility

Consumer concerns regarding the unfeasibility of IVM [20,28,34] is evident in the literature. There are concerns on the feasibility of IVM affordability and technology. Many consumers perceived IVM to be an expensive product due to its high manufacturing costs. Some opponents argue that it is unfeasible to call IVM an 'animal-free' meat because the culture medium may contain foetal bovine serum (FBS) or an animal-derived serum [28]. Researchers are still working on removing FBS because it is not a commercially viable option due to cost. Some consumers are unsure about the principles behind IVM technology and are sceptical of IVM feasibility to achieve "massive production from a single cell". Doubts and uncertainties regarding IVM feasibility among consumers could be a result of lack of familiarity and relevant information provision about IVM technology. However, these concerns can be reduced as IVM gains popularity in future.

### 3.2.3. Regulatory Efficiency

Due to the commercial unavailability of IVM products, there has been no regulatory framework governing IVM until recently in 2020, when Singapore became the first country to issue the first regulatory approval for lab-grown meat. For this reason, many consumers are worried about regulatory issues such as food quality [40]. As a plausible solution to these concerns, consumers demand that health, safety, and quality control checks to be

carried out on IVM before the product is available commercially. Concerns regarding IVM food safety tests, marketing strategy and labeling of products were raised by consumers to help with framing stringent regulations [41]. Similar reactions were observed in other studies, where consumers believed that an efficient regulatory system was essential for consumers to build their trust with IVM [36]. However, in November 2018, FDA and USDA along with the Good Food Institute issued a joint statement on regulations towards IVM [28]. This joint agreement could potentially resolve all the trust issues faced by consumers and help establish trust in the future.

### 3.2.4. Distrust in Technology

According to Bearth, "Trust is especially important in the absence of the possibility to judge the risk by oneself, which frequently is the case with new or complex food technologies" [42]. In the case of IVM, distrust is commonly observed when consumers reject IVM technology. Consumers concerns about IVM being introduced in the markets without their knowledge [42] indicated that people do not trust corporate food manufacturing companies. They believed that food companies are more concerned about making money and less concerned about their welfare. Distrust towards IVM can be attributed to consumers' lack of scientific knowledge, lack of trust in IVM technology, and lack of trust in researchers involved in IVM production. Furthermore, distrust in IVM also arises due to its unnaturalness. However, this can be resolved by improving IVM familiarity among consumers, and by creating an efficient regulatory system in the future [43].

### 3.3. Summary

In general, the consumer perceptions of IVM are still polarised. Understanding the consumers' sociodemographics, eating habits, income, and their view towards environment and public health, could assist in improving consumer perception of IVM. IVM is a new technology and still in its early commercialisation stage. Hence, many consumers have expressed concerns around the unnaturalness, safety, quality and feasibility of commercial production of IVM products. In addition, the lack of knowledge around IVM production has partially contributed to the distrust in IVM technology. Hence, a discussion on the current IVM production methods will be reviewed in the following sections.

## 4. General Background of IVM Muscle Formation

IVM works on the concept of myogenesis, a process of muscle formation. The formation of muscular tissue occurs during embryonic development, by multipotent myoblasts cells of mesodermal origin. These myoblast cells undergo embryonic fusion, proliferation and differentiation [44,45], to form myotubes, which gradually combine to forms muscle fibres as seen in Figure 1. The process of skeletal muscle formation is analogous to the production of IVM because a piece of meat is merely a mass of edible muscle fibres. The article published by Buckingham et.al in 2003 [39] has more detailed information on the process of muscle formation [46].

### 4.1. IVM History and Development

IVM is also known as 'cultured meat', 'lab-grown meat' and 'clean meat' [47,48]. It has also been referred to as 'animal-free meat', 'slaughter-free meat', 'vat meat' 'synthetic meat', 'artificial meat', 'shmeat', 'frankenmeat' and 'test-tube meat'. The Food and Drug Administration (FDA, Silver Spring, MD, USA), United States Drug Administration (USDA, Silver Spring, MD, USA) and other in vitro meat companies have jointly termed IVM as 'cell-based meat' [49] The most popular IVM term used is 'cultivated meat'.

The concept of IVM has been prevalent since Winston Churchill in 1931 predicted the future and stated that, "fifty years hence, we shall escape the absurdity of growing a whole chicken in order to eat the breast or wing, by growing these parts separately under a suitable medium" [50]. Research on in vitro cultivation of muscle fibres only began as early as 1971 when a researcher cultured immature aortal cells from guinea pig to obtain

myofibrils on eight weeks of culturing [51]. A few years later, another researcher cultured goldfish cells, which eventually developed into fish fillets [52]. During this period, a Dutch researcher turned entrepreneur, Willem Van Eelen received a patent for producing edible meat using collagen and muscle cells without killing animals [53]. In 2014, People for the Ethical Treatment of Animals (PETA), a US-based nonprofit organization, exhibited their support towards IVM by announcing a prize money of 1 million dollars to those who could cultivate lab-grown meat/in vitro meat (IVM) using chicken cells [54]. Besides that, the National Aeronautics and Space Administration (NASA) supported the idea of IVM since it can be used as a long-term food product for space missions [55]. IVM regained popularity as Dr Mark Post in 2013 created a beef burger patty using cow cells in a live event in London (BBC, 2013). Following this event, production of cell-based meat using IVM technology has created much attention in the past few years and is currently one of the most researched topics in food science.

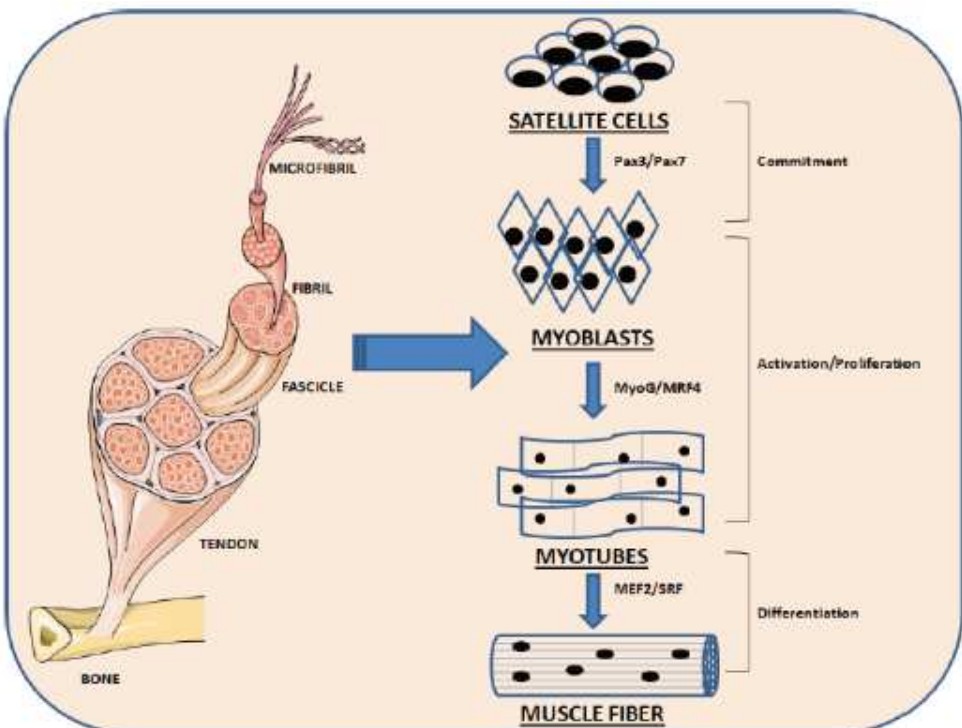

**Figure 1.** Schematic representation of muscle formation using skeletal muscle satellite cells, which undergoes proliferation and differentiation to produce a muscle fibre (Ultimo et al., 2018).

Currently, IVM research is supported by nonprofit organisations such as the Good Food Institute (GFI), and start-up companies such as New Harvest, CUBIQ Foods, Mosa Meat, Memphis Meats and JUST Inc. Furthermore, business tycoons such as Bill Gates, Sergey Brin and Richard Branson have invested in cell-based meat research and IVM start-up companies [56]. Hitherto, the number of investors in IVM and sustainable foods has increased exponentially. These investments have had a significant impact on the production of IVM as the costs for an IVM burger patty is now about $2400, which was $330,000 previously [57]. However, in 2018, IVM proponents hoped that the price will be as low as $5 by 2021 [58]. According to the Forbes website March 2022, the price of cell-cultured meat has decreased from $330,000 to about €9 or $9.80 per burger, which is now much more affordable.

The IVM industry is ever-increasing, and many start-up companies in the US and European countries now produce cell-based meats. In addition, even southeast Asian countries have extended their support towards IVM with China recently signing a 300-million-dollar deal to import slaughter-free meat from an Israel firm [59]. Japan has also struck a deal with JUST Inc. to deliver cell-based Wagyu beef in the market [60]. According

to industry experts, IVM will be commercially available in about five years due to huge technological advancement.

Negative Repercussions of Traditional Meat Industry and the Need for Meat Alternatives

The Food and Agriculture Organisation (FAO) predicted that the rate of meat consumption will increase by two thirds, i.e., about 73% in 2050 [61]. Such a tendency is expected to last for nearly four decades [62]. In other words, an increase in meat consumption levels will result in increased demand for meat products. This increase in meat production will result in increased adverse effects such as global climate changes, global warming, greenhouse gas emissions (GEG), increased carbon footprint and animal suffering. Besides, traditional meat production also causes water pollution due to the increased use of fertilisers and pesticides [63]. According to the United Nations (UN) and FAO, about 30% land usage, 6% of freshwater usage and 14.5% of greenhouse gas emissions (Figures 2 and 3) are linked to the livestock meat industry [62,64]. Moreover, New Zealand alone emits 49.2% of its total GHG emissions from the agricultural sector [65], which includes both the livestock and meat industry.

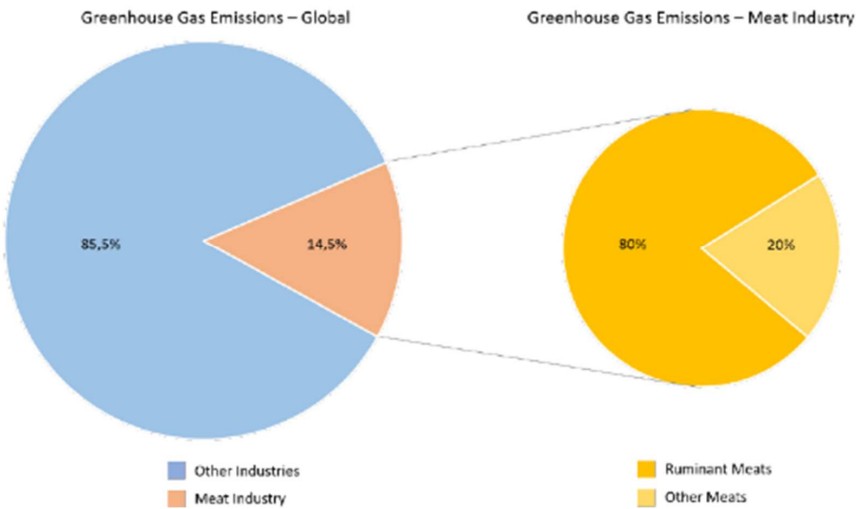

**Figure 2.** Greenhouse gas emission by the meat industry [27,33,66,67].

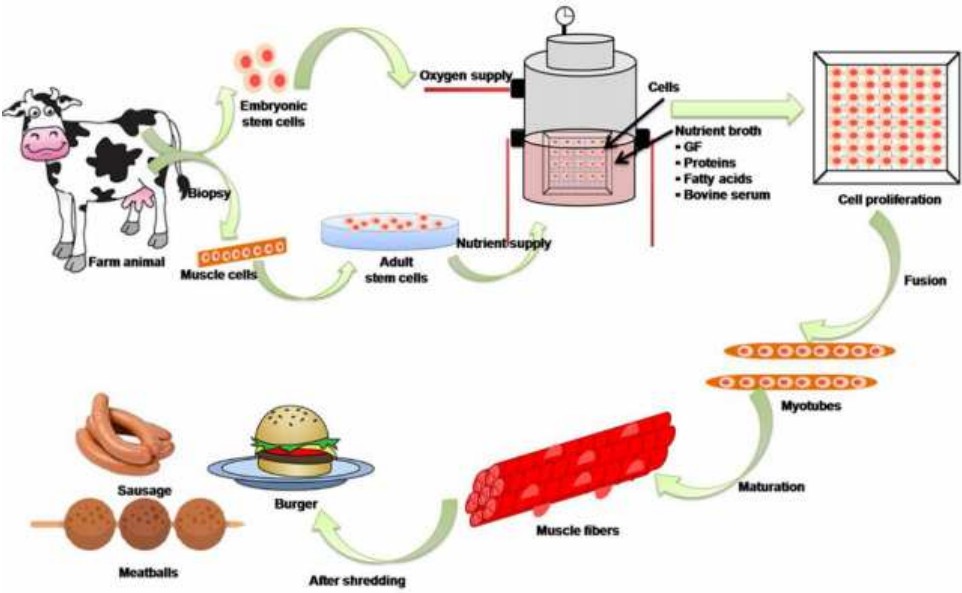

**Figure 3.** Schematic representation of IVM production (Gaydhane et al., 2018).

For all the reasons mentioned above, there is a dire need across the world for sustainable meat alternatives such as IVM to meet the global demand for meat. According to few researchers, IVM takes up 99% less land, 45% less energy, 96% fewer greenhouse gases emissions compared to the traditional meat industry [67–70]. The greenhouse gases (GHG) emissions of IVM are as low as traditional pork and poultry industries [70,71]. Furthermore, IVM is unlikely to cause any adverse environmental impacts and may in fact promote the reversal of climate changes [72].

## 5. IVM Production

In the last few years, many researchers have worked relentlessly to produce IVM efficiently. Previously muscles were grown in a petri dish [49,73], while recently others have incorporated various culture conditions, bioreactors, and various technologies such as scaffold method [74,75], self-organising method [76], and micropatterning [73]. Furthermore, other technologies such as organ printing, biophotonics, nanotechnology, and 3D printing are currently being considered for IVM Production [49]. Currently, IVM is produced in a bioreactor where the extraction, cultivation, and harvest of cells of interest is carried out for IVM production (Figures 2–4). The cells are later subjected to cell proliferation and differentiation stages in a bioreactor where muscle cells grow into muscle fibres. The desired cells are subjected to stimulations under optimum culture conditions, which facilitates cell growth. In most cases, the cells are grown on three-dimensional scaffolds as this provides highly structured meat-like steaks, whereas in some cases, the cells can self-organise to form muscle fibres. However, the self-organising technique does not offer structured meats but is suitable for processed meats like sausage, mince and patties.

**Figure 4.** Mechanism involved in self-organising tissue engineering technique [77].

A brief account on the cells, culture condition, stimulation of cell and other information on IVM production is described in Section 5.1. Table 2 summarizes general findings on conditions used for IVM production.

### 5.1. Cells and Culture Conditions

#### 5.1.1. Cells

There are various cell sources for IVM production, and these include adult stem cells (ASC), tissue-specific stem cells, mesenchymal stem cells (MSC), and induced pluripotent stem cells (iPSCs). The most widely used cells are embryonic stem cells and myosatellite cells [74,78].

**Table 2.** Cell sources and culture conditions for IVM production.

| Method | Cells | Bioreactor | Culture Medium | Presence of Growth Factor/Antibiotics | Culture Conditions | References |
|---|---|---|---|---|---|---|
| **Explant/BAMS** [1]/**co-culture** using perforated stainless-steel disks or cellulose acetate | Goldfish derived crude cell mixture and ATCC brown bullhead fibroblasts | N/A | 1. Minimal essential medium (MEM) [2] With Hanks salts [3] and Earle's salts [4] with foetal bovine serum (FBS). 2. 10% FBS [5] in 90% MEM in Hanks' salts. 3. Collagenase + incomplete MEM (minus FBS) Hanks' salts. 4. FBS substituted with dried Shiitake or Maitake mushrooms and fish meal extracts. | No antibiotics | Temperature-23 °C pH-7.2 | [79] |
| **Scaffold**-P(HEMA) [6], PGA [7], PLGA [8], PHB) [9] | Mc Coy Cell line—mouse endothelial cells | Rotating bioreactor-Synthecon RCCSID4 | Nutrient media | Growth factors and/or cytokines [10] | Cell cultured for 7 to 21 days in 37 °C/5% $CO_2$ incubator | [80] |
| **Scaffold**-Sodium alginate and fish gelatin type-1 with gelling agents such as agarose and glycerol | Myoblast cell line C2C12 | N/A | **Proliferation medium**: DMEM [11] media with 10% FBS + 2 mM L-glutamine, **Differentiation medium**: DMEM + 2% Horse serum | Antibiotics-penicillin and streptomycin | Cell cultured 37 °C and 5% $CO_2$ in a humidified atmosphere | [27] |
| **Scaffold**-alginate | C2C12 cells | N/A | 1. **C2C12 Growth medium**: DMEM high glucose—4.5 g/L with sodium pyruvate without L-glutamine and 10% FBS. 2. **C2C12 differentiation medium**: same as above but FBS replaced by Horse serum. 3. PBS | 1% of L-glutamine and 1% penicillin | Cell cultured in T-25 flasks at 310K with 5% $CO_2$ and 96% humidity. pH –7 to 7.3 At room temperature | [81] |
| Self-assembled tissues on 3D agarose-PDMS [12] scaffolds | Murine C2C12 myoblasts, human dermal fibroblasts, or TGF-1-differentiated myofibroblasts | N/A | **Growth media (GM)** consisting of DMEM with 10% FBS. **Differentiation medium**: DMEM, 2% horse serum | **GH**: 1 ng/mL of TGF-β1 (Transforming growth factor beta-1) **Antibiotics**: Penicillin and streptomycin | Cultured for 7 days with media replenishment every 2 to 3 days | [82] |

**Table 2.** *Cont.*

| Method | Cells | Bioreactor | Culture Medium | Presence of Growth Factor/Antibiotics | Culture Conditions | References |
|---|---|---|---|---|---|---|
| Scaffold | Muscle cell | Stirred tank bioreactor | Cyanobacteria hydrolysate | Growth factors and vitamins | Cell cultured for 60 days at 37 °C, at 100 rotations rpm aeration 0.05 vvm [13] | [83] |
| Scaffold-collagen spores | C2C12 cells of permanent myogenic cell line | N/A | **Basal medium**: Dulbecco's modified Eagle medium (DMEM) **For expansion**: above media + 10% foetal calf serum (FCS). **For differentiation**: fusion medium (FM)containing 2% FCS, 1% Insulin-Transferrin-Selenium-A, 670 µg/L sodium selenite, 11 g/L sodium pyruvate, 1 g/L insulin and 550 mg/L transferrin | Antibiotics, Penicillin and streptomycin sulphate. | Cultured on collagen spores in a tissue culture flask. Incubated in a humid chamber at 37 °C | [84] |
| Scaffold-collagen (surgisis) [14] | C2C12 myoblast cells Human Umbilical Vein Endothelial | N/A | 1. DMEM supplemented with 20% FBS, and 2.5% HEPES [15] buffer 2. Endothelial cell medium 3. DMEM supplemented with 10% FBS and 1% nonessential amino acids | VEGF (vascular endothelial growth factor) and FGF2 (fibroblast growth factor) | Constructs were placed in Krebs Henseleit [16] solution with 95% $O_2$, 5% $CO_2$ gas mixture at 25 °C | [2] |
| Scaffolds-Polydimethylsiloxane (PDMS) | Murine C2C12 myoblast cells | N/A | **-Growth medium**: containing DMEM + 10% foetal bovine serum (FBS) Matrigel was added for faster cell spreading, | 1%-penicillin and streptomycin | N/A | [85] |

[1]. BAMS—Bioartificial Muscle System. [2]. MEM—is a synthetic cell culture used to maintain cells in a tissue culture. [3]. Hank's salt—salt rich in bicarbonate ions, which are used as a buffer system in cell culture media and aid in maintaining the optimum physiological pH (roughly 7.0–7.4) for cellular growth. [4]. Earle's salt—a type of salt solution used for short–term maintenance of cells in a $CO_2$ environment. [5]. FBS—Foetal Bovine Serum. [6]. PHEMA – Polyhydroxyethylmethacrylate. [7]. PGA—Polyglycolide or poly glycolic acid. [8]. PLGA—Poly(lacto-co-glycolic acid). [9]. PHB—Polyhydroxybutyrate. [10]. Cytokines are groups of proteins, peptides or glycoproteins that are secreted by specific cells of the immune system. [11]. DMEM—Dulbecco's modified Eagle medium, a basal medium used for supporting the growth of cells. [12]. PDMS—Polydimethylsiloxane. [13]. Vvm—volume of air per unit of medium per unit of time. [14]. Surgisis—a bio-mesh with high collagen content used in biomedical processes. [15]. HEPES—(4-(2-hydroxyethyl)-1-piperazineethanesulfonic acid). [16]. Krebs Henseleit solution is a nutrient-rich solution (with sodium, chloride, calcium, magnesium sulfate, bicarbonate, phosphate, glucose, albumin and tromethamine) which is used in ex vivo studies.

Myosatellite Cells

Satellite cells are the most preferred cell sources to produce IVM skeletal muscle tissue. Satellite cells are also known as myoblast cells, myosatellite cells or muscle stem cells, which belong to the adult stem cells (ASC) category. It exhibits multipotency and is analogous to embryonic myoblast cells [49]. Satellite cells are mononucleated adult stem cells (ASC) situated at the periphery of skeletal muscle myofiber. When the myofibers are damaged in an injury, the satellite cells become activated, divide, and fuse to replace the damaged myofibers. These myofibers eventually form myofibrils and muscle tissues at later stages. Similar effects will be observed if these cells are used in IVM production. Hence satellite cells are the most preferred cell source.

The literature supports that satellite, myoblast, or adult stem cells are the most suitable type of cells due to their high regeneration power [86,87], ability to replicate myogenesis [74,88], and capacity to produce mature cells with specialised morphologies and functions, unlike the embryonic stem cells (ESCs). Hence, these cells are widely used in IVM research [79], especially the C2C12 myoblast cells [82,83,89–92], and in bio-artificial muscle production (BAMS) [93,94]. Moreover, adult stem cells (ASCs) such as epithelial cells have been used previously in the in vitro muscle production system [78]. However these cells require stimulation to form myoblasts, which may result in faulty myogenesis [95]. Besides that, myosatellite, myoblast or ASC's have a few drawbacks such as limited proliferation rate and susceptibility to turning cancerous, if cultured for extended periods.

Embryonic Stem Cells (ESC)

Stem cells that are derived from the embryo are known as embryonic stem cells (ESC) [96], which are either extracted from bovine or porcine. These are preferred cell sources for IVM production [97] due to their pluripotency and unlimited self-renewal capacity. For this reason, some researchers claim that IVM produced using ESC could provide sufficient meat to feed global hunger [42]. Thus, for the above reasons, ESC is considered a potentially good cell source for IVM production. However, there is currently no availability of bovine, ovine or porcine-derived ESC, with only murine cell lineages available. Hence ESC has to be differentiated into myogenic progenitor cells (MPCs) before muscle fibres can be formed. Besides that, other concerns of ESC in IVM production are its requirements of cell differentiation to produce myoblasts [42,95,98]. These stimulated cells are susceptible to loss of their existing proliferative characteristics at any later stages, despite culturing [99]. In addition, there is difficulty in maintaining undifferentiated embryonic stem cells [80,100], as opposed to other cell sources for IVM production.

Induced Pluripotent Stem Cells (iPSC's)

Induced pluripotent stem cells are differentiated cells, which are already transfected to induce pluripotency in cells. These iPSC's can be used as a cell source for IVM production [49,101] due to their myogenic differentiation capacity and injury repair mechanism [102]. However, there is no scientific evidence regarding their usage in IVM production. Only one study used fibroblast cell co-cultured with goldfish explant in the production of a bio-artificial muscle system (BAMs) [103]. On a different note, iPSC's are often co-cultured with fat cells such as adipose tissue-derived stem cell (ADC) to improve the texture, flavour and tenderness of IVM by increasing the amount of intramuscular fat [104]. These co-culturing techniques have been observed in other studies [89,105,106].

Dedifferentiated Cells

Dedifferentiated cells are the cells that have been reversed from terminally differentiated cells into multipotent cells, such as mature adipocytes. These cells on dedifferentiation give rise to multipotent preadipocyte cell line known as dedifferentiated fat cells (DFAT) [106]. In addition, these cells produce skeletal myocytes (muscle cells) when transdifferentiated [89]. Dedifferentiated cell properties make it suitable as a cell source for IVM production. However, some researchers argue that terminally differentiated cell proper-

ties of transdifferentiation, dedifferentiation and multipotency may be unusual features exhibited by cell-like substances, rather than the cell [81,82,107].

Despite the variety of cell sources available, IVM production is still a challenging task due to chances of cell death. However, some studies suggest that death can be prevented either by using immortal cells or by immortalisation of cell lineage. Furthermore, another challenge is to create an environment (in vitro) that mimics in vivo conditions for the optimum growth of cells. Detailed information on culture conditions is covered in Section 5.1.2.

### 5.1.2. Culture Conditions

Typically, culture conditions include factors such as culture media, serums, growth hormones and parameters such as pH, temperature, oxygen potential, pressure, and mechanical/electromagnetic/gravitational simulations for cells to produce IVM. However, defining a range, as well as optimising and stabilising the above parameters, is a challenging task, during large scale IVM production.

### Culture Medium

The culture medium plays a vital role in IVM production because it serves as a nutrient source for the growth of cells. The culture medium must be simple, edible optimal, affordable, and readily absorbable because it is used in considerable amounts. Traditionally, natural medium such as blood plasma was used for animal cell culture [108], but currently, researchers work with inexpensive serum-free medium such as Dulbecco's Modified Eagle Medium (DMEM) [74,82,83,94,109] and Ultroser G, which has all the essential nutrients, growth factors, adhesion factors and binding proteins [74]. However, the disadvantage of an animal-friendly and serum-free medium is the cost involved. Nevertheless, few researchers suggest that culture medium with both plant-based extracts and partially purified growth factors are economical and beneficial for the growth of cells [83]. Besides, the requirement for culture medium changes depending on cell growth or stages of development, such as differentiation and proliferation media used. Thus, there is a growing need for a continuous supply of edible and animal-friendly medium, which facilitates the growth of cells.

### Serum

Traditionally, animal-based serums such as foetal bovine serum (FBS) [81,110,111], foetal calf serum (FCS) [49], fishmeal extract and horse serums were used. However, these serums are neither ethical nor economical to use in the long run. Furthermore, animal-based serums have a high risk of carrying pathogens [96]. Researchers are now working with animal-free serum such as algal (cyanobacteria) or fungal/mushroom-based (Shiitake and Maitake mushrooms) [76], which makes it animal-friendly in nature. Interestingly, these serum-free medium have higher growth rates compared to FBS [95], due to sphingosine 1-phosphate and amino-acids in mushrooms [82,96].

### Growth Factors

Growth factors are essential components for the production of IVM [112], but the formulation and optimisation of growth factors is a challenging task [113] because it is dependent on the type of cells used. The commonly used growth factors for IVM production include Transforming Growth Factor-$\beta$ (TGF-$\beta$), Fibroblast Growth Factors (FGFs), Insulin-like Growth Factors (IGF) VEGF (vascular endothelial growth factor), and FGF2 (fibroblast growth factor) [114]. However, TGF-$\beta$ decreases the myoblast recruitment and differentiation [115], whereas FGFs are more stimulatory in nature compared to TGF—$\beta$. FGFs can enhance the rate of myoblast proliferation and prevent differentiation [116]. Similar to FGFs, a splice variant of IGF-1, called the mechano growth factor, increases proliferation of myoblasts [117] and differentiation in C2C12 myoblasts [118].

### 5.2. Processes Involved in IVM Production

### 5.2.1. Self-Organising Technique

The self-organising technique is a scaffoldless tissue engineering technique, which allows cells or tissues to grow freely using external forces such as physical manipulation or thermal input. Additionally, self-organisation works on the principle of tissue fusion as seen in Figures 2–5. Tissue fusion is a process in the developmental biology stage, where two or more identical tissues meet and fuse to form a continuous structure [119,120]. The tissues produced using the self-organising technique have native tissue morphology and can be grown up to several centimetres [49,90,121].

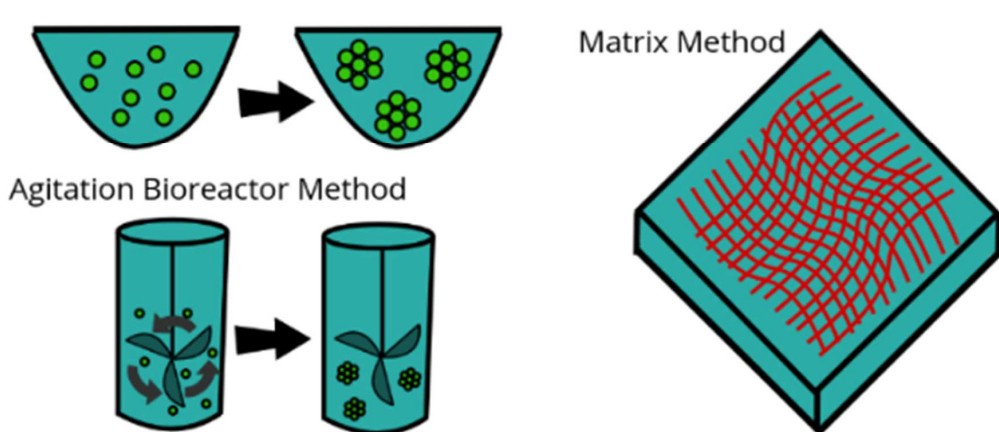

**Figure 5.** Scaffold free technique and scaffold techniques in tissue engineering, which can be used in IVM production (source: Declan Riordan [CC BY-SA 4.0 (https://creativecommons.org/licenses/by-sa/4.0)], https://commons.wikimedia.org/wiki/File:3d_cell_culture_(1).svg accessed on 19 April 2022).

The self-organising technique was first reported by Benjaminson et al. (2002), who produced the first-ever in vitro meat using goldfish (*Carassius auratus*) skeletal muscle explant, which regenerated and rearranged itself without any matrix, and showed a growth rate of 79% from the initial state. The explant was identical to the goldfish skeletal muscle and resembled fish fillets in terms of appearance and odour [75].

The self-organising technique is beneficial as it mimics skeletal muscles by retaining all the tissues which form the meat. Another benefit is its ability to produce a highly structured meat, unlike other methods. However, there are significant drawbacks with this method such as its susceptibility to undergo necrosis in the absence of blood supply [122]. Furthermore, IVM produced by this method requires the need for in vivo blood supply or vascularisation and an excretory mechanism to expel metabolic waste [78]. A major drawback of the self-organising technique is its inability to produce highly structured meat because it produces nonstructured and soft-consistency meat, which is only suitable for sausages, minced meat and burger patties [74].

### 5.2.2. Scaffold Technique

> "*A scaffold or matrix for a tissue engineering product refers to the ability to perform as a substrate that will support the appropriate cellular activity, including the facilitation of molecular and* mechanical signalling *systems, in order to optimize tissue regeneration, without eliciting any undesirable local or systemic response in the eventual host*". [95]

The scaffold technique is a tissue engineering technique, which uses three-dimensional structures made of hydrogels as a substrate to grow cells or tissues of interest. This method has gained immense popularity in the last few years and is widely used in IVM research [123,124] because scaffolds can act as a matrix for cell adherence to produce edible IVM skeletal muscle tissues. The scaffolds are made from hydrogels of natural or synthetic polymers, which are designed according to requirements and are later seeded with cells of interest. These cell-laden scaffolds are immersed in nutrient-rich medium contained in a bioreactor. Under favourable conditions, these cells grow into myotubes, which eventually form myofibrils. On maturation of cells, the resultant muscle fibres are harvested as edible IVM skeletal muscle tissue. Scaffolds developed using natural and edible hydrogels such as collagen and gelatin can produce complex meat with 3D structures. Thus, collagen and gelatin-based scaffolds are widely used to grow skeletal muscle tissues.

### 5.2.3. Bioreactors

The production of IVM can be carried out either using scaffolds or by scaffold-free techniques, such as the hanging drop method or agitation bioreactor method as seen in Figure 5.

Bioreactors, as seen in Figure 6, are large enclosed stainless steel units used for culturing cells in a sterile manner, and they provide a favourable environment for the proliferation of cells [27,125]. The production of IVM is facilitated by biophysical factors such as agitation and shear that are achieved by the inclusion of bioreactors. Bioreactors are generally equipped with a media source, scaffolding system, oxygenation system and a plumbing system for the continuous inflow of media and outflow of metabolic wastes and recycled media. Here, the cells are either suspended freely or seeded onto a scaffold suspended in a bioreactor. The cells then undergo proliferation and differentiation to yield 3D muscle fibres, which can be potentially used as IVM. The bioreactor can help with IVM production in several ways. First, bioreactors help with the continuous suspension of culture media, so the cell culture is not deprived of the nutrient source. Second, it helps with agitating using a low shear so that the suspended tissues are unaffected. Third, it assists with adequate oxygen perfusion as oxygen gradient influences the cell viability and density. Fourth, it assists with continuous contraction of cells, which eventually undergoes differentiation to produce myofibers.

There are several types of bioreactors that can be used for IVM production. This includes the rotating wall/vessel [76,126], stir tank [127], direct perfusion [128], rotatory [129], hollow fibre [124,130], wave mixed [131], rotatory bed [78], parallel plate [132], fixed bed [133] and Synthecon [134] reactors. However, a stir tank bioreactor is the most used in IVM production [135–138].

### 5.2.4. Stimulation of Cells

In an in vivo system, all the cellular process occurs naturally. These processes are carried out by nerve stimulation and electrical transmission/stimulation, with the help of an extracellular matrix (ECM). However, this is not the case when cells are grown in vitro. The challenge lies in mimicking the in vitro environment like the in vivo environment. Thus, in vitro systems require external stimulation of cells, which is brought about in two ways, either by electrical or mechanical stimulation as described in the following sections.

Electrical Stimulation

The electrical stimulations mimic the nerve stimulations, which assist with the formation of highly differentiated and functional skeletal muscle tissue. Typically, electrical stimulation in in vitro studies is carried out by passing an electrical stimuli via salt bridges, which are dispensed in culture media [139]. The general set up of electrical stimulation units is shown in Figures 2–8. However, there are a few setbacks with this method, such as a limited working area, making it difficult to work with various cell types at a time. Besides, the media-bridge system is susceptible to temperature fluctuations and exchange

of salts and ions during electrical stimulation of cells can result in alteration of temperature, pH and salinity. Furthermore, the electrical stimulation system is incapable of running multiple chambers making it difficult to maintain sterility [93].

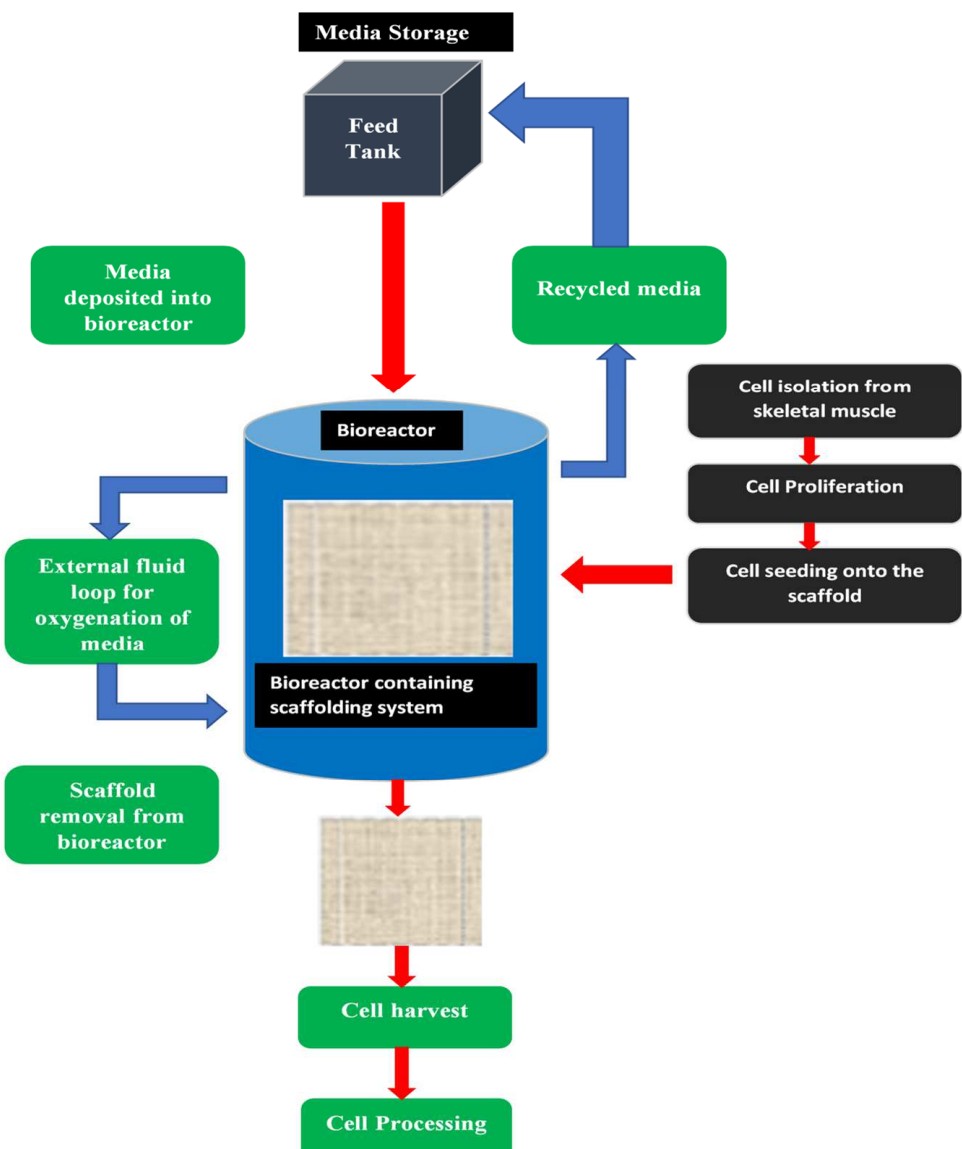

**Figure 6.** Schematic representation of IVM production using a bioreactor (Gaydhane et al., 2018).

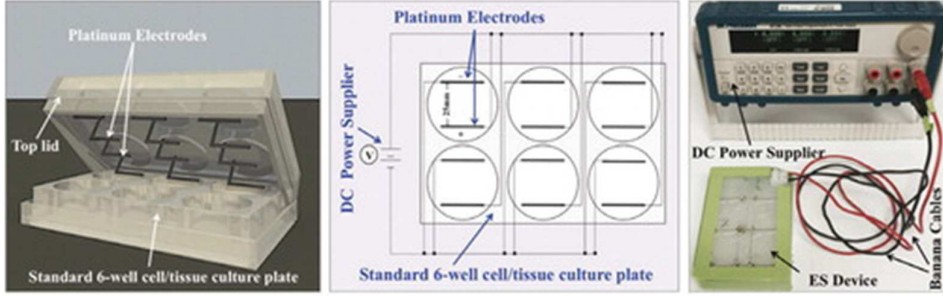

**Figure 7.** Set up of electrical stimulation, which is applied to cells to facilitate cell proliferation [95].

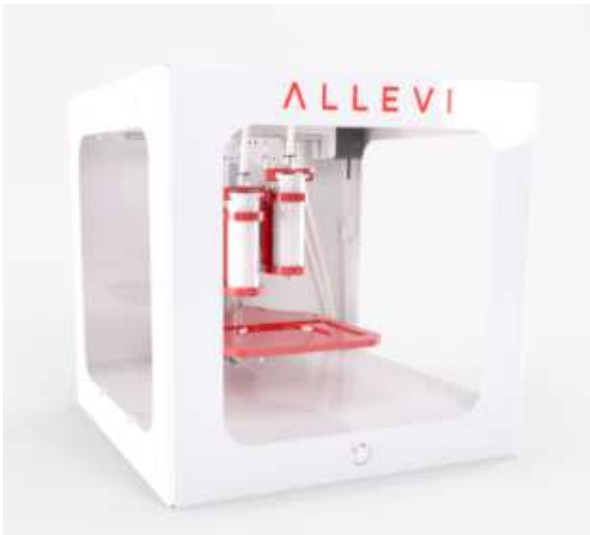

**Figure 8.** The Allevi 2 3D bioprinter that can be used to print GelMA, collagen hyaluronic acid scaffolds [140].

Electrical stimulation of cells during its growth is critical for IVM production for several reasons. First, electrical stimulation induces neuronal activity in the mature muscle fibres [141] and can be carried out by applying an electrical stimuli [142]. Second, electrical stimulation helps in accelerating the maturation of myotubules to develop early cross striations in C2C12 murine myoblasts. Third, electrical stimulation helps with neuronal activity by initiating contraction and differentiation of myotubules to eventually form myofibres [143].

Mechanical Stimulation

Mechanical stimulation is a biophysical stimulus that can be provided during myogenesis [42] as it influences gene expression, protein synthesis and total RNA/DNA content. Furthermore, it also helps with myofibre composition, cell number and muscle fibre diameter [91]. Mechanical stimulation of cells also helps with IVM production in several ways. First, it helps by applying mechanotransduction (a process through which cell sense and respond to mechanical stimuli by converting them to biochemical signals that elicit specific cellular responses), which alters the cell proliferation and differentiation rates [144]. Second, it helps with the fusion, alignment, and maturation of the myotube. Third, it helps with the proliferation and differentiation of muscle cells [145], muscle alignment [146] and muscle growth and maturation [147,148]. There are various methods of mechanical stimulation of cells [149]. The mechanical force is generated by using a perfusion bioreactor. When these mechanical forces are applied, this leads to perturbations in muscular protein conformation. This results in the exposure of hidden binding sites, which indirectly increases the signaling process in cells entrapped in the scaffold [150,151]. However, mechanical overloading can result in deformation, remodeling of cell, and can even affect cellular functions [152].

5.2.5. 3D Bioprinting

3D bioprinting is a novel method to create three-dimensional scaffolds of different hydrogel compositions. There are different types of printers available based on the technology, such as laser-assisted printing [152], as well as extrusion-based and inkjet-based 3D printing [151]. These 3D printers can efficiently create complex shapes of scaffolds with high resolution [153]. However, extrusion-based 3D printing is the most commonly used 3D printer [154]. The two main methods of 3D printing involve the use of either cellular scaffolds with cell-laden bio-ink or acellular scaffolds with hydrogels [155]. There are several studies on 3D printings for tissue engineering and regenerative medicine pur-

poses [74,78,156–158], but not many on the 3D printing skeletal muscle cells for meat purposes [158,159].

3D Printing of Scaffolds

Extruded scaffolds can be developed using the Allevi 2 bioprinter, as seen in Figure 8. It is a Fused Deposition Modeling (FDM) bioprinter that runs on a compressed air pneumatic system [160]. It has two extruders, where the first extruder extrudes bio-inks, and the other extruder is used for photocuring (visible or UV light) extruded scaffolds. The hydrogel or bio-ink is dispensed in syringes with needles of 0.3 mm nozzle diameter to extrude scaffolds of synthetic polymers. The bioprinter works on a three-dimensional computer-aided design (CAD) software such as Slicr and Repertoire host, which helps in designing scaffolds. The 3D CAD models of the desired scaffolds are sliced into 2D cross-sections, to adjust printing parameters such as speed, in-fill density, gauges, nozzle diameter, print temperature, number of layers, layer height and air pressure. Software such as the Slicer or Repetier Host combines the two-dimensional cross sectionals of the scaffold to form a computer-aided three-dimensional structure. The bioprinter has a triaxial system (x, y, z), which allows scaffolds to be printed into desired shapes, which are then cured using built-in UV light to carry out crosslinking reactions.

Hydrogels

Hydrogels are three-dimensional polymer gels, which are made up of water-soluble polymers that are held together by water-insoluble cross-linkages. In other words, hydrogels are formed by the crosslinking of homopolymers or copolymers to give 3D structures with unique mechanical and chemical characteristics. Generally, most hydrogels swell and increase their weight when added to water due to their imbibition property. These hydrogels can be further classified into chemical or physical hydrogels based on the crosslinking mechanisms [83,161,162]. Physical hydrogels are not permanent, whereas chemical hydrogels are permanent. Hydrogels are either natural or synthetic in nature, and the natural hydrogels are comprised of polysaccharides and proteins and are usually found in the tissues such as agarose, gelatin, elastin, alginate, cellulose chitosan, fibrin, collagen and Matrigel [108,163] and hyaluronic acid. Natural hydrogels are more preferred due to their extracellular matrix-like structure, which enables cell growth, solute transport, cell binding and other cellular behaviours [83].

Synthetic hydrogels are usually made of polyethene glycol (PEG) [164], poly-(L-lactic acid) (PLLA) and polylactic-glycolic acid (PLGA) [165], polydimethylsiloxane (PDMS) [166], and poly(vinyl alcohol) (PVA) [160]. Semi-synthetic hydrogels such as gelatin methacrylate/gelatin methacryloyl (GelMA) have been used previously to culture cells [167]. Moreover, these synthetic hydrogels can be used in the production of highly structured 3D IVM because they can facilitate cell entrapment and cell growth. Hydrogels, in general, are extensively used in biomedical sciences applications [42,162,168,169] because they mimic extracellular matrix [170] and due to their biocompatibility, biodegradability, density and crosslinking properties. Moreover, these hydrogels offer a promising approach for skeletal muscle tissue engineering for many reasons. First, they allow dense cell entrapment uniformly in the hydrogel scaffold [171]. Second, they assist with myotube alignment due to the in vivo like environment. The major drawback with hydrogels is their instability, but this can be managed by co-culturing cells that produce extracellular matrix, which stabilises the matrix while the hydrogel degenerates [172]. The reproducibility and uniformity of the gels can be adjusted by electrospinning, but this may result in non-uniform distribution of cells.

Crosslinking Reaction

In chemistry, cross-links are referred to as bonds that connect one polymer chain to another through covalent bonds or ionic bonds. These cross-linkages are either formed by covalent [173,174] or noncovalent interactions [174,175] to form either chemical gels or

physical gels, respectively. The process of crosslinking can be carried out either physically or chemically. Chemical crosslinking is carried out by either polymerization, chain-growth polymerization, sulphur vulcanization or by chemical reactions such as addition and condensation irradiation. It can also be performed by irradiation using high energy x-ray, electron beam and gamma rays. On the other hand, physical crosslinking is conducted by ionic interactions, crystallization, stereo complex formation, and protein interaction. Crosslinking is vital because it affects physicochemical properties of polymers such as elasticity, viscosity, swelling, solubility and strength of gels [176]. A detailed description of crosslinking of polymers is described in Ahadian et al. (2015) [162].

Polymerisation Reaction

Polymerisation is a crosslinking method where several monomer (homopolymers or copolymers) units react together chemically to form three dimensional polymers. Polymerisation is essential for hydrogels as it determines the physicochemical properties based on its monomers. A detailed description of the polymerisation reaction is beyond is described by Levental et al. (2009).

Types of Hydrogels

Collagen Hydrogel

Collagen is a naturally occurring protein that makes up 25% of the protein content in mammals [177]. Besides, it is the main component of extracellular matrix [178], which provides an in vivo-like environment that enables cell encapsulation, cell binding and integrin signalling [177]. In addition, collagen also provides exceptional crosslinking ability [162], low antigenicity, biodegradability [179] and higher biocompatibility [180]. Therefore, collagen hydrogels are widely used as a scaffold in tissue engineering [162]. However, they have a few drawbacks. First, they are soft and susceptible to degradation, but this can be managed either by increasing the amount of collagen or by chemically modifying it to prevent degradation [177]. Second, they can trigger an immune response occasionally, which can affect cell culture. Third, the usage of collagen hydrogels, in the long run, is not an economical option [181]. Finally, collagen demerits include thermal instability, low mechanical strength and susceptibility to contaminations [162,182].

Gelatin Hydrogels

Gelatin is a natural polymer, which is obtained by collagen hydrolysis. It is an economical, temperature-responsive polymer with high cell adhesiveness [183]. In addition, gelatin hydrogels are often functionalized with a cross-linkable component like methacryloyl group, which is crosslinked by photoinitiators [184] to enhance hydrogels stability [165]. However, most gelatin hydrogels are prone to hydrolysis, but this can be managed by chemical modification [185].

Gelatin methacryloyl (gelMA) is a modified form of gelatin, which is obtained by chemical crosslinking. It is one of the most widely used hydrogels because it offers excellent biocompatibility, physicochemical properties, printability and is cost effective [179,186]. Consequently, GelMA has been widely used in cell culture studies as an extracellular matrix due to its exceptional cell binding characteristics [187], as well as cell migration, differentiation and proliferation properties [188]. Nevertheless, there are a few demerits such as low mechanical strength [165,189,190], as well as reduced cell distribution and migration [191]. However, this can be managed by combining gelMA with hyaluronic acid or and silk fibroin [192].

Hyaluronic Acid Hydrogels

Hyaluronic acid (HA) or hyaluronan is a polysaccharide abundantly found in connective, epithelial and neural tissues. It is responsible for the formation of extracellular matrices with the help of glycosaminoglycans. Furthermore, it also helps with cell proliferation, cell migration, and other cellular functions. However, HA requires purification before hydrogel preparation to eliminate impurities and toxins. Hyaluronic acid has been

used in many genetic engineering applications [190–192] because it is biocompatible with cells and mimics in vivo conditions. In addition, it helps with angiogenesis in engineered tissues [193] to promote vascularisation. Thus, HA has been widely used in the biomedical and tissue engineering research for over 30 years [193].

## 6. Conclusions

This review adds to a growing body of knowledge in the literature to further understand the perception of IVM among consumers and its method of production. Consumers were found to have mixed responses towards IVM products. Acceptance of IVM was dependent not only on consumer sociodemographics and eating habits, but it also extends to political affiliation and beliefs towards the technology. Interestingly, there is some degree of distrust from consumers towards IVM products stemming from its lack of regulatory efficiency and a sense of control. Despite the distrust, the IVM industry continues to grow with big players based in the US and European countries due to the shifting paradigm in consumers' perception of sustainability. Conceptually IVM can be produced through myogenesis where cells join to create muscle fibres using various techniques described in this review. For efficient production of IVM, it is important to consider source of cells, cell culture conditions, media, and type of culture technique. With increasing consumers awareness of sustainable consumption of food, there is a need to improve IVM technology to produce affordable, high quality IVM. Like many other new technologies, the consumer's attitude towards IVM may shift with increased awareness and availability of IVM. Future research should focus on identifying ways of improving consumers acceptance of IVM. In addition, it would be critical to improve efficiency of IVM production to yield affordable, acceptable and high quality IVM product.

**Author Contributions:** Conceptualization, K.K., N.H., M.M.M., Y.L., A.S.; methodology, K.K., N.H., Y.L., T.L., A.S.; software, K.K., T.L.; validation, M.M.M., Y.L., T.L.; formal analysis, K.K., N.H., M.M.M., T.L.; investigation, M.M.M., Y.L.; resources, N.H., A.S.; data curation, K.K.; writing—original draft preparation, K.K., N.H., M.M.M., Y.L., T.L., A.S.; writing—review and editing, K.K., N.H., Y.L., T.L., A.S.; visualization, M.M.M., T.L.; supervision, N.H., A.S.; project administration, K.K.; funding acquisition, N.H., A.S. All authors have read and agreed to the published version of the manuscript.

**Funding:** This research received no external funding.

**Institutional Review Board Statement:** Not applicable for studies not involving humans or animals.

**Informed Consent Statement:** Not applicable for studies not involving humans.

**Data Availability Statement:** Data sharing not applicable. No new data were created or analyzed in this study. Data sharing is not applicable to this article.

**Conflicts of Interest:** The authors declare no conflict of interest.

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
