# Peer review of "Consumer Acceptance and Production of In Vitro Meat: A Review"

_sustainability, doi:10.3390/su14094910_

Round 1

Reviewer 1 Report

The manuscript is well written and critically analyzed the consumer acceptance and production aspects of in vitro meat. It will definitely have a good readership and citations. I have the following suggestions for enhancing its readership as

1) It is better if authors further expand the area of animal welfare and ethical/ religious concerns such as kosher and halal concerns with the inputs briefly.

2) A section on the recent development and setting of start-ups in the field of commercializing of in vitro meat production  

Author Response

Please see attached for detailed amendments.

Reviewer 2 Report

This paper reviews cultured meat, including consumer acceptance as well as technical aspects. The latter is beyond my expertise; the former is an adequate review, but needs better structure, and is not particularly novel. I recommend English language editing throughout, and have additional comments below.

  1. Introduction

L35: It is also worth acknowledging the benefits to animal welfare; e.g. currently over 90% of farmed animals are factory farmed.

L38: I don’t think it’s accurate to say that consumer acceptance is the most significant hurdle to overcome here; there are still several technological challenges to overcome.

L40: Taste is also a major factor not mentioned here

L45: Quantitative and empirical are not mutually exclusive – do you mean mostly qualitative rather than quantitative? Or mostly theoretical rather than empirical?

L71: Missing line?

  1. Methodology

More detail is needed in this section – e.g. exact search terms + databases, rather than only some of them. What were the inclusion/exclusion criteria?

  1. Consumer perception of IVM

L94: It is not clear how you have distinguished ‘consumer’, ‘personal’, and ‘ethical’ concerns – it seems like many of the examples could fit into any category

L150: I think this discrepancy is due to Hocquette’s study design: first, they surveyed people they knew personally, many of whom were working in the meat industry! They also had poor question design, with answer options that were not mutually exclusive for the key measure.

L155: In fact the majority of research suggests that meat-eaters are more likely to eat IVM vs. vegetarians, and meat attachment is a predictor of acceptance.

L181: Compared to which countries? Source 16 actually shows USA is lower than China/India

L188: This contradicts the above section on lowest acceptance among vegetarians

L234: I don’t think this title fits here; the previous sections were on demographics associated with higher acceptance, whereas this section is on reasons for accepting.

L265: more precisely, it is not sustainable or scalable – we will simply run out of space

L269: this could also be seen as a benefit, if it enables hungry people to eat meat when they otherwise would not be able to

L303: it is worth noting that manufacturers are working to remove FBS, and it won’t be commercially viable with FBS included due to cost

L313: it is worth mentioning that Singapore has regulated cultured meat since 2020

L342: the transition to the next section is quite abrupt; I recommend a summary of section 3 and a brief prelude to section 4

4. General background of IVM muscle formation

L354: incomplete sentence

L362: the nomenclature discussion is far from settled, but the current most popularly used term is cultivated meat

L367: New Harvest is a nonprofit, not a startup

L395: by 2021? Update

L430: you can cite the much more recent and comprehensive life cycle analysis from CE Delft - Odegard & Sinke 2021

L445: this section seems like it should be part of the intro; many of the points about the benefits of CM are repeated from above 

5. IVM production

This section is beyond my expertise; I trust that other reviewers can provide helpful comments here

6. Conclusion

It would be helpful to highlight priority areas for future research

Author Response

(The authors gave the same response as above.)

Round 2

Reviewer 2 Report

The authors have done a good job of addressing my concerns. 

I am happy to endorse the paper for publication following extensive English language editing.

Author Response

We'd like to thank the reviewer for this and have amended the manuscript accordingly

This manuscript is a resubmission of an earlier submission. The following is a list of the peer review reports and author responses from that submission.

Round 1

Reviewer 1 Report

The present review draft is very well written and compiled available recent literature with respect to the production and consumer acceptance aspects of in-vitro meat. It will definitely have good viewership and citations. The second section of the draft is very well compiled with quality and informative figures. I have most of the observations related to the first section ie regarding consumer acceptance such as 

1) Authors used some abbreviations without their proper full version such as IVM, ESC; please check and correct

2. I feel the abstract needs major editing and at present form, it is not as per the standard format of journal such as paragraphs, references, and word count; please only write the significant background, methods, and findings.

3. Line 46-55; these concerns are also affected by the product types, production process, and labeling; please add these also in the draft to improve their impact

4. Line 142: please check the sentence as India and China; especially in India, meat attachment is not appreciable. Also, other factors such as environmental and animal welfare concerns also affect this.

5. Line 174 onwards; As while discussing consumer acceptance of IVM, the issue of animal welfare and their potential sustainability issue are very critical; so as per my opinion, please add more information/ further elaborate on these topics for making the draft more valuable and complete

6. Please add some more information about the status of vegetarian/ vegan aspects of IVM based on culture media component and also religious issue such as not using any cells from pigs for Muslims etc as these issues will affect consumer acceptance. 

7. Line 311-318; plz mention the recent companies started the production of IVM as EAT JUST in Singapore and Rohovat city in Israel by future meat technologies; Ivy Farm Technology plan for IV pork

8. line 320-322: the increasing demand will mainly come from developing countries whereas in developed countries are even now making efforts to decrease it; please add these aspects also

9. Line 348: Suffrage?? Authors may replace it with a more suitable word